# Cost-Driven Design of Printed Wideband Antennas with Reduced Silver Ink Consumption for the Internet of Things

**DOI:** 10.3390/s22207929

**Published:** 2022-10-18

**Authors:** Nicolas Claus, Jo Verhaevert, Hendrik Rogier

**Affiliations:** Department of Information Technology, Ghent University-imec, Technologiepark-Zwijnaarde 126, 9052 Ghent, Belgium

**Keywords:** flexible antenna, ink reduction, Internet of Things (IoT), meshed antenna, nanoparticle silver ink, printed antenna, screen printing, wideband antenna, 5G

## Abstract

The Internet of Things (IoT) accelerates the need for compact, lightweight and low-cost antennas combining wideband operation with a high integration potential. Although screen printing is excellently suited for manufacturing conformal antennas on a flexible substrate, its application is typically limited due to the expensive nature of conductive inks. This paper investigates how the production cost of a flexible coplanar waveguide (CPW)-fed planar monopole antenna can be reduced by exploiting a mesh-based method for limiting ink consumption. Prototypes with mesh grids of different line widths and densities were screen-printed on a polyethylene terephthalate (PET) foil using silver-based nanoparticle ink. Smaller line widths decrease antenna gain and efficiency, while denser mesh grids better approximate unmeshed antenna behavior, albeit at the expense of greater ink consumption. A meshed prototype of 34.76×58.03mm with almost 80% ink reduction compared to an unmeshed counterpart is presented. It is capable of providing wideband coverage in the IMT/LTE-1/n1 (1.92–2.17 GHz), LTE-40/n40 (2.3–2.4 GHz), 2.45 GHz ISM (2.4–2.4835 GHz), IMT-E/LTE-7/n7 (2.5–2.69 GHz), and n78 5G (3.3–3.8 GHz) frequency bands. It exhibits a peak radiation efficiency above 90% and a metallized surface area of 2.46 cm^2^ (yielding an ink-to-total-surface ratio of 12.2%).

## 1. Introduction

The Internet of Things (IoT) aims at supplementing our daily lives with an additional layer of ubiquitous intelligence. By giving ordinary objects the capabilities of identification, sensing, computation, and communication, they can be transformed into intelligent devices [1]. As a result, the number of connected devices has been exponentially rising over the past few years, spurring important developments and new applications in a variety of markets, including smart wearables, Industry 4.0, and home automation.

In IoT systems, wireless communication plays a key role in providing interaction between the user and the device, between devices themselves, and in transmitting data for business, analytics, or monitoring purposes [2]. However, due to the current lack of standardization and the multitude of IoT applications, wireless transceivers must operate within a wide range of communication protocols and frequencies in order to guarantee maximal interoperability. Frequencies for IoT can be in the sub-6 GHz spectrum or the mmWave spectrum [3,4], and are predominantly concentrated around the 2.45 GHz industrial, scientific and medical (ISM) band [5]. Hence, antenna systems are required to exhibit wideband or multiband operation within traditional IoT standards such as WLAN, Bluetooth, WiMAX, or Zigbee, as well as in cellular standards such as 3G, 4G, and future bands for 5G (e.g., the 3.3–3.8 GHz band) [6].

In addition to the need to operate across a wide range of frequencies, antenna design for IoT devices faces other inherent challenges related to size, material choice, and cost. Recently, embedded or conformal antennas have been adopted to fulfill all of these requirements. When antennas are considered as an integral part of the IoT device, more integrated designs with smaller form factors are possible. Additionally, because part of the antenna is constructed using the already available and often low-cost materials of the IoT device itself, this is a more cost-efficient and durable approach than conventional antenna design, where the antenna is regarded as a stand-alone supplementary component.

An interesting trend in this regard is research that concentrates on printed antennas for conformal applications [7,8]. Common low-cost materials, such as paper [9,10,11,12], cardboard [13], textiles [14], and plastics [15,16] may be used as a printable substrate. Moreover, additive printing processes such as inkjet printing, screen printing, flexography, and gravure printing are considered simple, fast, and cost-efficient compared to traditional subtractive manufacturing technologies [17,18,19,20]. Due to its maturity and good reliability, screen printing is currently regarded as the dominant printing technology [9].

Contemporary research mainly focuses on the use of new substrates [5] or on the development of novel highly conductive inks, which are based on organometallic particles, carbon nanotubes, or silver nanoparticles [21,22]. These inks are generally very expensive, and as a result the overall manufacturing cost of a printed antenna is typically dominated by the amount of required metallization. Hence, in order to facilitate cost-effective mass manufacturing for the IoT, antenna designs with low ink consumption are preferred.

In the literature, several printed wideband or multiband antennas have already been presented. In [23], an inkjet-printed circular-shaped monopole antenna with ultra-wideband (UWB) characteristics was proposed for IoT applications. It is printed on a polyethylene terephthalate (PET) substrate and operates within the 3.04–10.70 GHz and 15.18–18 GHz frequency bands. Abutarboush et al. [24] have presented a screen-printed wideband antenna on a flexible Kapton substrate, with a defected ground structure (DGS) used to enhance the impedance bandwidth. In [25], a wideband screen-printed antenna using a graphene flake-based ink was proposed. However, all these designs require a significant amount of ink compared to the overall surface area, rendering them not very cost-effective for mass IoT production.

In order to reduce the amount of ink required for a given antenna design, different strategies have been explored by other researchers. In [26], the authors investigated whether printing with a variable ink layer thickness, that is, using a thicker ink deposition at positions with a higher current density, could reduce ink consumption without sacrificing antenna performance. However, it turned out that the radiation efficiency mainly depends on the total amount of ink utilization, rather than on its distribution. Alternatively, the application of mesh strategies has been studied extensively for both printed and non-printed antennas. In [27], the radiation performance of common patch antennas was investigated for varying mesh grid densities. However, despite showing promising results in terms of ink reduction potential, these antennas were designed for radio frequency identification (RFID) applications, and as a result lacked wideband or multiband operability in keeping with the currently envisaged IoT and 5G standards. A single-layer antenna consisting of proximity-fed meshed patches was presented in [28], exhibiting good performance with high gain and dual-band operation. However, it was implemented on a rigid substrate and suffered from low fractional bandwidth.

In other works [29,30,31,32], meshed antennas have been primarily reported for their optically transparent properties rather than for ink reduction. In [29], a metal mesh film (MMF) was employed on a PET foil to implement transparent antennas for ultra-high definition (UHD) TV applications. In [30], the trade-off between antenna transmission characteristics and optical transparency was analyzed for different mesh patterns consisting of square- and diamond-shaped mesh cells. Alternatively, the optical transmittance of meshed structures has been exploited to enable efficient integration of antennas with solar energy harvesters [31,32]. However, most of these studies do not take into account practical printing processes and the associated limitations for the mesh optimization. Furthermore, because they focus on the conventional trade-off between antenna efficiency and optical transparency, the optimization of the antenna’s mesh geometry is often considered from a one-sided perspective without considering the solution’s cost-effectiveness.

In this paper, a flexible passive wideband antenna is presented for operation in the IMT/LTE-1/n1 (1.92–2.17 GHz), LTE-40/n40 (2.3–2.4 GHz), 2.45 GHz ISM (2.4–2.4835 GHz), IMT-E/LTE-7/n7 (2.5–2.69 GHz), and n78 5G (3.3–3.8 GHz) frequency bands. The antenna is screen-printed on PET foil using a high-performance silver nanoparticle-based ink. To substantially reduce the overall cost related to ink consumption, a suitable mesh strategy is exploited. Moreover, specific mesh properties such as line width and grid density are analyzed in order to explore the trade-off between antenna performance and ink reduction. To the best of the authors’ knowledge, this is the first time that the design and optimization of a mesh-based antenna has been regarded primarily from a cost-driven perspective while ensuring good multiband antenna performance for modern IoT applications. The proposed antennas are excellently suited for such applications thanks to their multi-standard operability, compact footprint, high radiation efficiency, and compatibility with additive manufacturing techniques.

The rest of this manuscript is organized as follows. The materials and methods for this research are discussed in the following section, with particular attention paid to the material parameters of the conductive ink and the fully electromagnetically characterized substrate. The design requirements are provided in Section 3, along with the proposed topology for the antenna in both meshed and unmeshed form. Section 4 presents a comprehensive analysis of the results, and Section 5 summarizes the key findings.

## 2. Materials and Methods

This section discusses the utilized materials and methods for the conducted research. Section 2.1 reports the properties of the conductive nanoparticle silver ink, while Section 2.2 describes the characterization procedure for the antenna substrate and the resulting electromagnetic parameters. Finally, Section 2.3 and Section 2.4 provide details on the manufacturing process and the electromagnetic modeling of the thin ink layer, respectively.

### 2.1. Nanoparticle Silver Ink

The novel high-quality nanoparticle silver ink used in this paper was ’PRELECT SI-P2000’, produced by Agfa-Gevaert in Mortsel, Belgium (URL: https://www.agfa.com). This ink contains tiny silver particles that enable more effective sintering compared to conventional flake-based inks. The final conductivity depends on both the sintering temperature and the time; generally, higher conductivity levels can be reached than with a flake-based ink. As a result, thinner ink layers may be printed, which is advantageous in terms of ink consumption and the associated production costs.

The high performance of this ink (which was formerly called ’Orgacon SI-P2000’) for printed conductors has already been verified in [9]; measurements on a PET foil after thermal sintering for 10 min in a 150 °C oven showed that it exhibits a very low DC sheet resistance (below 5 mΩ/sq). Furthermore, a volume resistivity below 3 mΩ/sq/mil (i.e., for an ink thickness of 1mil=25.4 μm) and a silver content around 70 Ag wt% (percentage by weight) have been specified by the manufacturer.

### 2.2. Substrate Characterization

A 125 μm-thick PET foil was used as a substrate. This material was first characterized using a two-step characterization procedure to accurately determine the electromagnetic parameters. Several test structures were produced and measured during each step.

In a first step, a nonresonant method was employed to calculate an initial estimate for both the effective permittivity and the loss tangent. More specifically, a modified matrix-pencil two-line method, which removes perturbations due to geometric uncertainties and imperfect connector de-embedding, was applied [33]. This method is based on the measurement of the transmission characteristic for two transmission lines of different lengths $l_{1}=19\mathrm{mm}$ and $l_{2}=61\mathrm{mm}$, as visualized in Figure 1a.

Figure 2 shows the characterized effective permittivity $\varepsilon_{r,\mathrm{eff}}$ and dielectric loss tangent tan(δ). This calculation is performed for 801 frequency points between 2 GHz and 6 GHz. The arithmetic mean for $\varepsilon_{r,\mathrm{eff}}$ and tan(δ) equals 1.38 and 0.054, respectively. Based on analytical expressions for the calculation of the effective permittivity for a conventional coplanar waveguide (CPW) line with finite ground planes on a substrate with finite thickness [34], the relative dielectric constant is estimated as $\varepsilon_{r}\approx 3.36$.

In a second step, the initial estimates resulting from the matrix-pencil two-line method were refined by a resonant method that relies on a straight coplanar resonator (CPR) [35]. By matching the measured S-parameters to the simulated resonances by varying both $\varepsilon_{r}$ and tan(δ) in CST Microwave Studio’s frequency domain (FD) electromagnetic field simulator, more accurate values for both parameters were determined. Two CPRs of lengths $L_{1}=59.6\mathrm{mm}$ and $L_{2}=119.6\mathrm{mm}$ were produced, as shown in Figure 1b. Tapering was used to employ the same connector landing pattern (for a Hirose U.FL-R-SMT-1 U.FL connector) in combination with wider lines, thereby mitigating high transmission losses. The simulations and measurements are shown in Figure 3.

Because the resonance frequencies of the CPR correspond to the resonant line’s length being (approximately) a multiple of half-a-wavelength [35,36], the longer resonator exhibits more resonance peaks than the shorter one. Therefore, it can be used to complement and fine-tune the results from the latter, especially over a broad frequency range. The resulting values for the relative dielectric constant and loss tangent are $\varepsilon_{r}=3.88$ and tan(δ)=0.055, respectively. These values are comparable with the results that were obtained with the first method, albeit more accurate.

### 2.3. Manufacturing Process

All samples were printed by Quad Industries in Sint-Niklaas, Belgium (URL: https://www.quad-ind.com/) using a screen printing process optimized for the aforementioned nanoparticle silver ink and PET substrate. Specifically, the fine line resolution was improved to more accurately print the meshed antenna topologies. A minimum track width of 100 μm was attained. The ink was cured for 5 min at 130 °C, and the final thickness of the ink layer deposited on the highly homogeneous PET foil was approximately 4 μm.

Reliable connectorization of printed samples on a flexible substrate is typically a challenging task; conventional soldering degrades the substrate, and conventional 3.5 mm SMA coaxial connectors are too heavy to provide a mechanically stable connection [2]. Therefore, a Hirose U.FL-R-SMT-1 U.FL connector was chosen as a lower-weight and more compact alternative. The connector was attached to the sample using conductive ink. After a curing step in the oven at 150 °C, the connection was mechanically strengthened by applying ultraviolet (UV) light-activated glue. Finally, the electrical connection between the connector and the metallized parts of the sample was verified.

### 2.4. Electromagnetic Modeling

Electromagnetic field simulations were performed with the FD solver of CST Microwave Studio’s simulation environment. The PET substrate was modeled using the characterized parameters, which were acquired as described in Section 2.2. However, because knowledge of the DC sheet resistance was insufficient to accurately characterize the ink’s electrical behavior at higher frequencies, the conductor was more challenging to model.

Based on the provided volume resistivity of 3mΩ/sq/mil, the ink was modeled as a thin homogeneous conductor with an electrical resistivity $\rho =3\;m\Omega \;\cdot \;25.4\;\mu m=7.62\times 10^{-8}\;\Omega m$, or equivalently, an electrical conductivity of $\sigma =1/\rho =13.12\times 10^{6}\;S/m$. For an ink layer with thickness τ=4 μm, the corresponding DC sheet resistance was calculated as $R_{s}^{\mathrm{DC}}=\rho /\tau =19.1\;m\Omega /\mathrm{sq}$. However, at higher frequencies the DC sheet resistance is inaccurate and should be replaced by a radio frequency (RF) sheet resistance $R_{s}^{\mathrm{RF}}$, as the skin effect becomes more significant. Then, the currents can no longer be considered as uniformly distributed over the entire cross-section of the conductor, such as in the DC regime; instead, they are concentrated along the conductor’s outer edges over a distance of the skin depth. Assuming a sufficiently large conductivity such that the dielectric and polarization effects can be neglected ($\sigma \gg 2\pi f\varepsilon_{0}\varepsilon_{r}$), the skin depth is approximated by
(1)$$\delta \approx \sqrt{\frac{\rho }{\pi f\mu_{0}\mu_{r}}},$$
with *f* the frequency, $\mu_{0}$ the permeability of free space, and $\mu_{r}$ the relative permeability of the conductor ($\mu_{r}=1$). In large, thin, and highly conductive metal planes and when neglecting proximity effects, there is only conduction in a bottom and top layer with a thickness corresponding to the skin depth δ, as justified by the exponential reduction in current density with depth in a conductor. Hence, the theoretical boundaries for the minimum RF sheet resistance are provided by
(2)$$R_{s,\min}^{\mathrm{RF}}=\left\{\begin{array}{cc} R_{s}^{\mathrm{DC}}, & f\leq f_{\tau =2\delta }\\ \frac{\rho }{2\delta }, & f\geq f_{\tau =2\delta } \end{array}\right.,$$
with $f_{\tau =2\delta }$ the frequency at which the skin depth equals half the thickness of the ink layer. Now, as a worst-case estimation, assuming that proximity effects are such that only one of these layers carries the major part of the current leads to a maximum RF sheet resistance equal to
(3)$$R_{s,\max}^{\mathrm{RF}}=\frac{\rho }{\delta }.$$

Combining Equations (Equation 1) and (Equation 3), Figure 4 depicts the calculated maximum RF sheet resistance from 2 GHz to 6 GHz. Its value equals 27.1 mΩ/sq at 2.45 GHz. Although this value is higher than the DC sheet resistance, it remains acceptable for realizing printed conductors.

It should be noted that while the theoretical study of the skin depth as presented above is helpful for quickly predicting the electrical conductance at RF frequencies, it is only a rough approximation of reality. Due to microscopic flaws such as microcracks and surface roughness, the thin ink layer actually behaves as a heterogeneous conductor. Nonetheless, it is treated as a homogeneous conductor in simulation, as it is very challenging to accurately quantize and model such heterogeneity. This approximation is acceptable considering the nominal high conductivity of the applied ink.

## 3. Antenna Design

This section discusses the design of the printed antennas, specifying the requirements in Section 3.1 and the antenna topologies in Section 3.2.

### 3.1. Design Requirements

The following design goals were formulated for the antenna design. First, the antenna size should be smaller than the format of a generic credit card, being 85.6 mm × 54 mm. Second, to guarantee high operability in a wide range of IoT protocols, a wideband antenna is desired, covering the IMT/LTE-1/n1 (1.92–2.17 GHz), LTE-40/n40 (2.3–2.4 GHz), 2.45 GHz ISM (2.4–2.4835 GHz), IMT-E/LTE-7/n7 (2.5–2.69 GHz), and n78 5G (3.3–3.8 GHz) frequency bands. It should be noted that the combination of these bands allows operation according to all relevant IoT standards, including WLAN, Bluetooth, WiMAX, and Zigbee, as well as in cellular standards such as 3G, 4G, and 5G. The magnitude of the reflection coefficient $|S_{11}|$ should be lower than −10 dB (with respect to 50 Ω) for all these frequencies. As a third requirement, a stable and omnidirectional radiation pattern is required. A high-gain highly directive antenna is not desired, because for IoT applications signals can impinge randomly from all directions, and a directional antenna is unable to pick up those signals without high amplitude differences. Nonetheless, a stable radiation pattern is required over the aforementioned frequency bands, with similar gain levels. Finally, in order to comply with IoT criteria for low-cost fabrication in high production volumes, the production cost should be minimized by reducing the ink consumption.

### 3.2. Antenna Topology

Based on the design criteria postulated in Section 3.1, a CPW-fed monopolar patch antenna was designed. CPW feeding was applied to excite the antenna, as it provides a much wider bandwidth than microstrip line feeding [37]. Moreover, because the ground plane is in the same plane as the signal trace and the antenna, it only requires single-layer metallization, making it compatible with the screen-printing process described in Section 2.3.

Section 3.2.1 focuses on the unmeshed monopolar patch antenna, which serves as a starting point for the meshed variant described in Section 3.2.2. The purpose of the meshing is to reduce the ink consumption and the associated manufacturing cost.

#### 3.2.1. Unmeshed Antenna

Figure 5 shows the topology of the unmeshed CPW-fed monopolar patch antenna along with the annotated design parameters. A footprint for a Hirose U.FL-R-SMT-1 U.FL connector was implemented in the ink layer, as shown magnified in the inset of Figure 5. The dimensions related to this connector footprint and the CPW feed line are the same for all manufactured samples, including the test structures of Section 2.2 and the proposed antenna prototypes, and are provided as follows: $F_{w}=2.20\mathrm{mm}$, $f_{w}=1.00\mathrm{mm}$, $P_{e}=0.60\mathrm{mm}$, $P_{l}=2.20\mathrm{mm}$, $P_{w}=0.35\mathrm{mm}$, s=0.45mm, and g=0.20mm.

Based on this topology, an antenna designated as Prototype I (P-I) was designed. The final dimensions for the parameters denoted in Figure 5, as optimized by the FD solver of CST Microwave Studio, are provided as follows: W=36.10mm, L=61.80mm, w=23.50mm, l=37.00mm, d=2.80mm, $G_{w}=16.75\mathrm{mm}$, and $G_{l}=12.00\mathrm{mm}$.

#### 3.2.2. Meshed Antenna

In order to reduce the ink consumption, the solid areas of the unmeshed antenna’s topology were converted into a meshed grid of conductive traces, resulting in the topology shown in Figure 6. Rectangular mesh cells were employed, to optimally conform to the antenna’s surface currents. Both the patch and the ground planes were meshed with an ($N_{p,x}\times N_{p,y}$) and ($N_{g,x}\times N_{g,y}$) mesh grid, respectively. For the planar monopole’s ground plane, the grid was symmetrically applied at both sides except for an exclusion zone of width *D*. The feed line was not meshed in order to ensure proper impedance matching for the CPW feed line and avoid mechanical instability after connectorization. The number of mesh cells in the *x*-direction is denoted by $N_{p,x}$ and $N_{g,x}$ for the patch and ground, respectively. Similarly, the number of mesh cells in the *y*-direction for the patch and ground is denoted by $N_{p,y}$ and $N_{g,y}$, respectively.

The size of the holes in the grid was different for the patch and ground plane due to a different number of mesh cells, resulting in hole dimensions of $w_{p}$ and $l_{p}$ for the planar monopole’s patch and $w_{g}$ and $l_{g}$ for its ground plane. A fixed mesh line width *t* was maintained. By comparing the surface areas of the non-metallized and metallized sections, the relative ink reduction compared to an unmeshed antenna of the same dimensions, denoted as ζ, can be calculated as follows:(4)$$\zeta =\frac{N_{p,x}N_{p,y}w_{p}l_{p}+2N_{g,x}N_{g,y}w_{g}l_{g}}{wl+2G_{w}G_{l}}.$$

Based on this topology, three prototypes with a different number of mesh cells for the patch were designed, denoted as Prototype II (P-II), Prototype III (P-III), and Prototype IV (P-IV). P-II was implemented using the coarsest mesh, while P-IV was based on the finest mesh for the monopolar patch. The prototypes all share the same global dimensions: W=34.76mm, L=58.03mm, w=23.28mm, l=36.55mm, d=2.98mm, $G_{w}=16.08\mathrm{mm}$, and $G_{l}=8.50\mathrm{mm}$. While keeping the ground plane’s mesh unchanged, the mesh grid for the patch itself was altered among the different prototypes. An overview of the mesh parameters is provided in Table 1.

## 4. Results and Discussion

In this section, the antenna topologies presented in Section 3 are examined with respect to the imposed design goals mentioned in Section 3.1. First, in Section 4.1, the influence of the adopted line width in the meshed designs is investigated theoretically through simulations. This allows the optimal mesh line width for production of the meshed antenna prototypes to be selected. Next, all fabricated antenna designs are measured and quantitatively compared in Section 4.2. This analysis primarily focuses on the trade-off between antenna performance and ink consumption by evaluating the behavior of the unmeshed antenna compared to the behavior of the prototypes with different mesh grid densities. Finally, the radiation performance of the unmeshed prototype and the best meshed prototype is experimentally validated in Section 4.3 and Section 4.4, respectively.

Measurements of the magnitude of the antennas’ reflection coefficients, $|S_{11}|$ (with respect to 50 Ω) were performed with a Keysight N9918A FieldFox Vector Network Analyzer (VNA). Furthermore, far-field radiation patterns were measured in an anechoic chamber using an NSI-MI spherical far-field antenna measurement system and a Keysight N5242A PNA-X VNA. The resulting data were obtained through gain comparison using an MI-12-1.7 and an MI-12-2.6 standard gain horn. The former operates from 1.7 GHz to 2.6 GHz, whereas the latter operates between 2.6 GHz and 3.95 GHz. Hence, using both permits the full frequency range of operation for the designed antennas to be covered.

### 4.1. Determination of Optimal Mesh Line Width

In a first step, the optimal line width for the meshed prototypes was determined through simulations. Therefore, an antenna with the dimensions of P-II (see Figure 6 and Table 1) was modeled in CST Microwave Studio’s simulation environment. With this model, simulations were performed using a mesh line width varying from t=100 μm to t=500 μm. The lower limit of 100 μm was imposed by the screen printing process; while samples with a line width of 50 μm were fabricated, they did not meet the expected quality requirements.

Figure 7 shows the simulated results. The magnitude of the reflection coefficient for various line widths is shown in Figure 7a. It can be observed that while the impedance bandwidth remains unchanged, the return loss between the two resonance peaks decreases for smaller line widths. The simulated radiation efficiency and peak gain are displayed in Figure 7b,c, respectively. The radiation efficiency decreases for smaller line widths, with this effect being more pronounced at higher frequencies. This can be attributed to the ohmic losses, which increase with conductor resistance. Printing the mesh with narrower traces causes more losses, as a conductor’s resistance is inversely proportional to its cross-sectional area. In conclusion, a trade-off is presented between the reduced amount of ink and the antenna performance, which includes impedance matching, radiation efficiency and antenna gain.

In order to reduce the ink consumption as much as possible, the mesh line width for the produced antennas was chosen as the lower limit imposed by the production process, that is, t=100 μm. As demonstrated in the analysis above, this slightly affects the antenna performance, although efficiencies better than 90% are obtained regardless.

### 4.2. Experimental Comparison of All Prototypes

In this section, all aforementioned antenna prototypes (P-I, P-II, P-III, and P-IV) are experimentally compared. The unmeshed antenna P-I was manufactured according to the dimensions indicated in the caption of Figure 5, while the meshed prototypes P-II, P-III, and P-IV were fabricated according to the dimensions in Table 1 and the caption of Figure 6. As noted in Section 4.1, a fixed line width equal to t=100 μm was applied for the antennas’ mesh grids. Figure 8 displays photographs of the fabricated meshed prototypes.

The simulated and measured reflection coefficients ($|S_{11}|$) of all antenna prototypes are shown in Figure 9. To provide more detail within the frequency region of interest, they are displayed both from 0 to 6 GHz (left subfigure) and from 1.5 to 4.5 GHz (right subfigure). For the unmeshed prototype P-I, the measured impedance bandwidth agrees very well with the simulated one, and all relevant frequency standards described in Section 3.1 are perfectly covered. They are indicated by colored boxes, with blue corresponding to IMT/LTE-1/n1 (1.92–2.17 GHz), orange to LTE-40/n40 (2.3–2.4 GHz), yellow to 2.45 GHz ISM (2.4–2.4835 GHz), purple to IMT-E/LTE-7/n7 (2.5–2.69 GHz), and green to the n78 5G frequency band (3.3–3.8 GHz).

Figure 9 shows a comparison between the prototypes with different mesh grids, that is, P-II, P-III, and P-IV. As predicted by the simulations, a coarser mesh results in a lower return loss between the two dips. This is comparable to the results presented in Figure 7a, as the experienced impedance for the electric currents increases for both smaller line widths and a lower number of current paths. However, the overall measured reflection coefficients are higher than in the simulations, violating the −10 dB matching criterion and resulting in dual-band behavior for P-II and P-III.

The higher mismatch can mainly be attributed to dimensional inaccuracies in the printed ink pattern; these result in a deviation in the input impedance compared to the simulated value of 50 Ω. The CPW feed region is particularly susceptible to this, as a very small gap *g* is employed between the signal line and the ground planes. Figure 10 shows microscopic images of this region for P-I (Figure 10a) and P-II (Figure 10b) as well as for a mesh line of P-II (Figure 10c). A corrugated pattern can be seen around straight edges in the ink layer, resulting in the aforementioned dimensional variation and an altered input impedance.

Because a denser mesh increases the return loss within the covered impedance bandwidth, the mismatch problem is resolved for P-IV. As seen in Figure 9, it spans the entire bandwidth of interest and covers all relevant frequency bands. Hence, only P-I and P-IV comply with the design requirements specified in Section 3.1.

Figure 11 shows the simulated surface currents at 2.45 GHz for the unmeshed antenna and the meshed variants. Figure 11a reveals a high current density around the CPW feed line. As previously mentioned, a mesh exclusion zone of width *D* is maintained for this reason. Because the patch’s current primarily flows in the *y*-direction, it is reasonable to assume that the antenna performance will improve as there are more current paths in this direction (i.e., for a higher value of $N_{p,x}$). However, $N_{p,y}$ is increased in this study as well because it enhances the mechanical stability of the manufactured sample and the unrestricted current flow necessary for the patch to radiate properly [38]. Comparing Figure 11b–d, it can be seen that for a denser mesh grid, that is, larger values of $N_{p,x}$ and $N_{p,y}$, the current flows along significantly more paths. Hence, the surface currents of the unmeshed antenna are better reproduced, leading to more similar antenna performance.

In Table 2, the fabricated antennas are compared in terms of compactness, absolute utilized ink surface area, and covered frequencies. For the meshed antennas (P-II, P-III, and P-IV), the relative ink reduction ζ compared to an unmeshed antenna of the same dimensions is specified, as calculated by Equation (Equation 4). The covered frequencies are the measured frequencies for which the −10 dB matching criterion is met. Multiple rows indicate dual-band behavior, as observed for prototypes P-II and P-III. As shown in Table 2, all antennas comply with the required size constraints. While P-I is slightly larger than the other prototypes, it covers all relevant frequency bands. However, ink is required for more than half of the total surface area, making this antenna quite expensive to produce. When the meshing technique is exploited, the ink consumption, and hence the cost, can be drastically reduced. Prototypes P-II, P-III, and P-IV all have an ink-to-total-surface ratio of around 10%. Furthermore, their relative ink reduction ζ varies between 85.8% and 79.8% from the most coarse to the most dense mesh grid, respectively. Hence, depending on the application, a choice can be made in the trade-off between antenna performance and ink consumption.

Due to their dual-band behavior, P-II and P-III do not comply with the specified design requirements in Section 3.1. Therefore, the following sections considering the antennas’ radiation performance examine only P-I and P-IV.

### 4.3. Radiation Performance of Prototype I (P-I)

Figure 12 shows the simulated (solid curves) and measured (dashed curves) radiation patterns for P-I at the center frequencies of the IMT/LTE-1/n1, 2.45 GHz ISM, and n78 5G frequency bands, that is, 2.05 GHz, 2.45 GHz, and 3.55 GHz, respectively. Because the antenna’s omnidirectional radiation pattern is difficult to measure with the current anechoic measurement setup, the radiation patterns in the rear hemisphere are negatively impacted by the presence of the antenna under test (AUT) positioner of the NSI-MI measurement system. This causes undesired scattering and interference effects, resulting in distorted and imprecise measurement data in this hemisphere. For example, the null at 180° is caused by the AUT positioner, which obstructs the line-of-sight between the measured antenna and standard gain horn. To indicate the unreliability of the measured results around 180°, this region is shaded in gray. The measurements obtained in the antenna’s frontal hemisphere, however, are very accurate, as there are no impediments there. Nonetheless, a slight ripple is noticeable as a result of small multipath reflections caused by the cables and the AUT positioner behind the antenna. In order to remove these effects and to decrease the ripple’s amplitude as much as possible, special care was taken to cover the cables and the antenna’s surroundings with RF-absorbing materials.

As demonstrated by Figure 12, the measured radiation patterns are in excellent agreement with the simulated ones for both the azimuth and the elevation plane. At lower frequencies, the boresight gain is slightly lower than predicted by simulation, whereas for higher frequencies the measured and simulated patterns correspond very well. Due to the antenna’s symmetry in the XZ-plane, a symmetrical and quasi-omnidirectional radiation pattern is obtained, as shown on the left side of Figure 12. However, the asymmetry in the YZ-plane introduced by the feed line and the ground plane results in a pattern that is slightly skewed at higher frequencies. Nonetheless, considering the wide frequency span, the antenna’s radiation pattern can be considered stable over the entire frequency range.

The measured and simulated peak gains as a function of frequency are shown in Figure 13 along with the simulated radiation efficiencies. The measured peak gain in Figure 13a is compensated for by the ripple observed in the measured radiation patterns due to scattering by the measurement setup, in particular the feed cable. This is achieved by simulating the antenna both in ideal conditions and in non-ideal anechoic circumstances, which are accurately modeled by carefully incorporating the measurement setup applied during the actual measurements in CST Microwave Studio. By comparing the simulation data for both setups, the ripple’s magnitude can be estimated and subtracted from the measured peak gain values. As for Figure 13b, in light of the excellent agreement between the simulated and measured radiation patterns, it can be assumed that the actual radiation efficiency is well approximated by the simulated radiation efficiency. The high efficiency is mainly due to the antenna design and fabrication; by employing a very thin substrate and adopting single-layer metallization with CPW feeding, the electromagnetic fields are predominantly concentrated in the air. Hence, the dielectric loss resulting from interactions of these fields with the substrate are minimized. Furthermore, the ink’s excellent conductive properties, as discussed in Section 2.4, yield low conduction losses.

### 4.4. Radiation Performance of Prototype IV (P-IV)

The simulated (solid curves) and measured (dashed curves) radiation patterns for the best meshed antenna, which was P-IV, are shown in Figure 14. Gray shading is again applied around 180° to indicate the negative impact of the measurement setup on the results in this region. A similar radiation performance is attained when compared to the unmeshed antenna P-I, with the measurements and simulations being in excellent agreement, especially at higher frequencies. This shows that P-IV, the meshed antenna with the densest grid, accurately approximates the behavior of P-I while requiring considerably less ink (corresponding to a relative ink reduction of ζ=79.8% compared to an unmeshed counterpart of the same dimensions; see Equation (Equation 4)).

The simulated and measured peak gain and the simulated radiation efficiency for P-IV are shown in Figure 15. Similar to Section 4.3, the measured peak gain in Figure 15a is compensated for by the ripple observed in the measured radiation patterns. The simulated radiation efficiency (displayed in Figure 15b) again provides a good estimation of the actual radiation efficiency owing to the agreement between the simulated and measured radiation patterns. As for the unmeshed antenna P-I, the high efficiency is mainly caused by low dielectric losses resulting from the thin substrate and CPW feeding. Moreover, by adopting a dense mesh that is specifically tailored to the antenna shape and simulated surface current profiles (as shown in Figure 11), additional conduction losses arising from increased line impedance are minimized. As a result, excellent agreement between the radiation properties of P-I and P-IV is achieved.

A comparison between P-IV and state-of-the-art printed wideband antennas is presented in Table 3. It can be seen that our solution consumes drastically less ink in both absolute and relative terms. Hence, the manufacturing cost is considerably lower.

## 5. Conclusions

In this contribution, a mesh-based method was investigated to reduce the manufacturing cost of a flexible screen-printed antenna by limiting the consumption of expensive silver-based nanoparticle ink. First, a compact CPW-fed monopolar patch was designed for wideband coverage of the IMT/LTE-1/n1 (1.92–2.17 GHz), LTE-40/n40 (2.3–2.4 GHz), 2.45 GHz ISM (2.4–2.4835 GHz), IMT-E/LTE-7/n7 (2.5–2.69 GHz), and n78 5G (3.3–3.8 GHz) frequency bands. Next, the antenna’s conductive area was converted into a meshed grid of conductive traces while considering the surface currents and mechanical stability to devise the optimal grid geometry.

The trade-off between antenna performance and ink consumption was investigated theoretically and experimentally for antennas with different mesh configurations. First, the line width’s influence on the radiation properties was determined through simulations. Due to increased ohmic losses, smaller lines were shown to reduce the antenna’s gain and efficiency. At the same time, a lower limit for the line width is usually imposed by the technical limitations of the printing process. Next, the effect of the mesh density was examined through fabrication of antenna prototypes with varying grid densities, resulting in various levels of ink reduction. A general comparison in terms of impedance matching revealed that the antenna with the densest mesh exhibited a higher return loss and yielded better similarity with the unmeshed antenna. A broad impedance bandwidth from 1.88 to 4.10 GHz (74.2%) was covered, supporting all required frequency bands. Finally, this design was compared to the unmeshed antenna in terms of radiation properties. It offered comparable radiation performance with similar gain and efficiency while consuming considerably less ink. More specifically, compared to its unmeshed counterpart with the same dimensions, the ultimate meshed antenna achieved an ink reduction of 79.8%.

The mesh-based antennas presented in this paper were designed and optimized based on a novel perspective, considering cost reduction as the main objective rather than optical transparency. The resulting meshed antenna is ideally suited for deployment in IoT applications thanks to its multi-standard operability (including both cellular and non-cellular standards) within a compact footprint, its high efficiency, and its compatibility with mature additive manufacturing technologies. Moreover, as the ink consumption is considerably reduced compared to the unmeshed antenna while achieving similar radiation characteristics, it is a very attractive candidate for cost-effective mass-production.

## Figures and Tables

**Figure 1 sensors-22-07929-f001:**
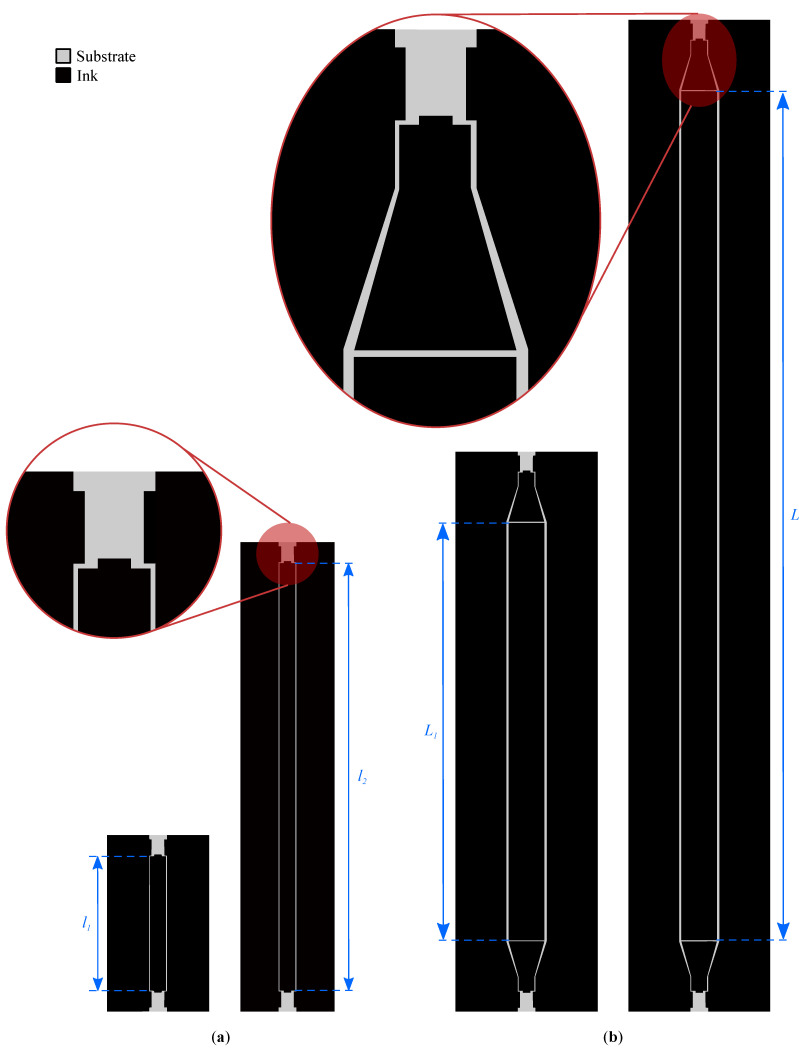
(**a**) Test structures for the first (nonresonant) step in the characterization procedure: coplanar waveguide (CPW) transmission lines of length $l_{1}=19\mathrm{mm}$ and $l_{2}=61\mathrm{mm}$. (**b**) Test structures for the second (resonant) step in the characterization procedure: tapered coplanar resonators (CPRs) of length $L_{1}=59.6\mathrm{mm}$ and $L_{2}=119.6\mathrm{mm}$. All test structures are displayed on the same scale. The added insets provide more detail about the feed sections.

**Figure 2 sensors-22-07929-f002:**
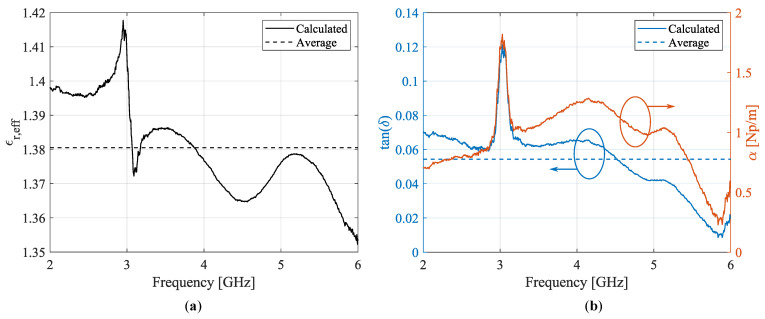
Calculated (solid lines) and averaged (dashed lines) parameters for the polyethylene terephthalate (PET) substrate resulting from the first (nonresonant) characterization step: (**a**) effective permittivity and (**b**) loss tangent, as well as the real part of the complex propagation factor γ=α+jβ.

**Figure 3 sensors-22-07929-f003:**
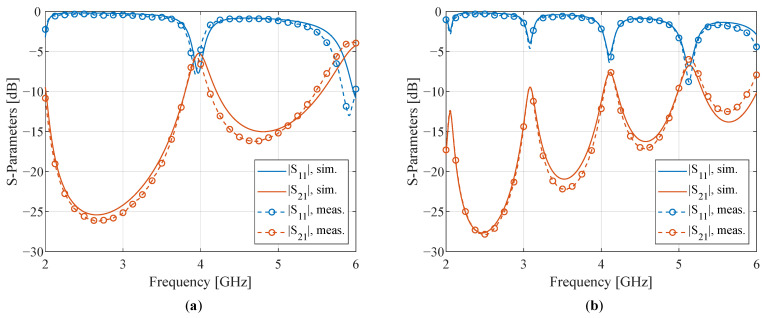
Mapping between simulation and measurement for (**a**) the short resonator, with length $L_{1}$, and (**b**) the long resonator, with length $L_{2}$. The resulting values for the relative dielectric constant and loss tangent are $\varepsilon_{r}=3.88$ and tan(δ)=0.055, respectively.

**Figure 4 sensors-22-07929-f004:**
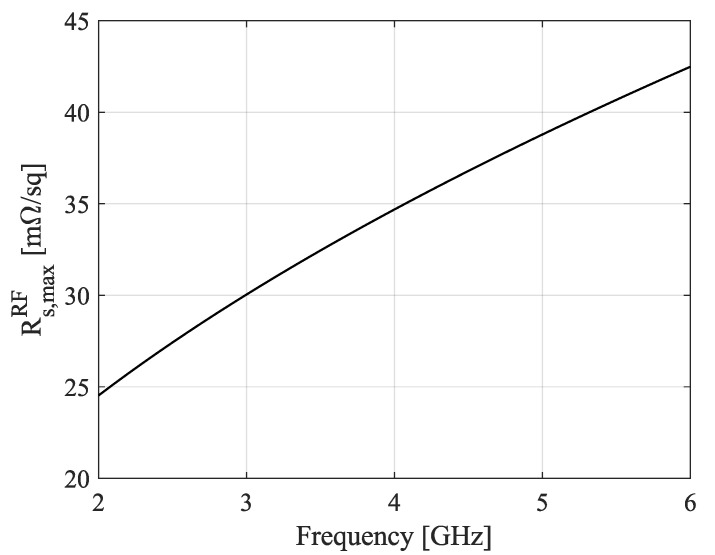
Theoretically calculated maximum radio frequency (RF) sheet resistance for the utilized ink layer.

**Figure 5 sensors-22-07929-f005:**
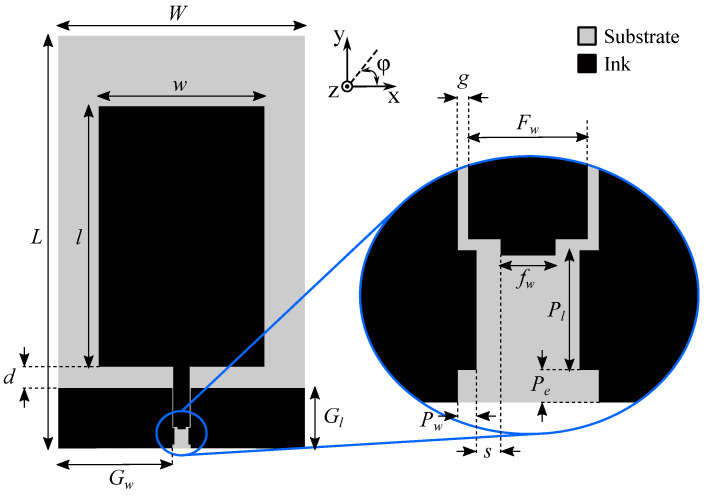
Topology of the unmeshed CPW-fed monopolar patch antenna. The inset shows more detail of the Hirose U.FL-R-SMT-1 U.FL connector footprint. The dimensions of Prototype I (P-I) are W=36.10mm, L=61.80mm, w=23.50mm, l=37.00mm, d=2.80mm, $G_{w}=16.75\mathrm{mm}$, and $G_{l}=12.00\mathrm{mm}$. The dimensions of the U.FL connector’s footprint are $F_{w}=2.20\mathrm{mm}$, $f_{w}=1.00\mathrm{mm}$, $P_{e}=0.60\mathrm{mm}$, $P_{l}=2.20\mathrm{mm}$, $P_{w}=0.35\mathrm{mm}$, s=0.45mm, and g=0.20mm.

**Figure 6 sensors-22-07929-f006:**
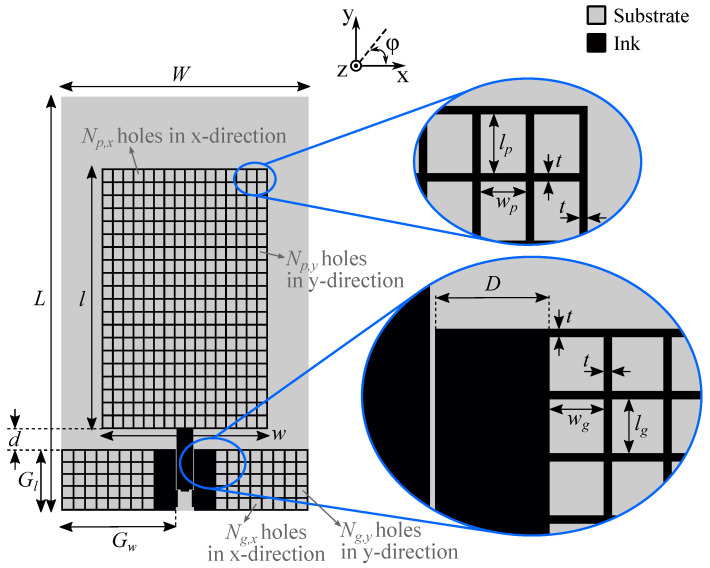
Topology of the meshed CPW-fed monopolar patch antenna, with a mesh grid ($N_{p,x}\times N_{p,y}$) and ($N_{g,x}\times N_{g,y}$) for the patch and planar monopole’s ground plane, respectively. The insets show more detailed dimensions of the mesh grid. Three prototypes were designed, denoted as Prototype II (P-II), Prototype III (P-III), and Prototype IV (P-IV). They share the same global dimensions: W=34.76mm, L=58.03mm, w=23.28mm, l=36.55mm, d=2.98mm, $G_{w}=16.08\mathrm{mm}$, and $G_{l}=8.50\mathrm{mm}$. The parameters of their mesh grids are different, as indicated in Table 1.

**Figure 7 sensors-22-07929-f007:**
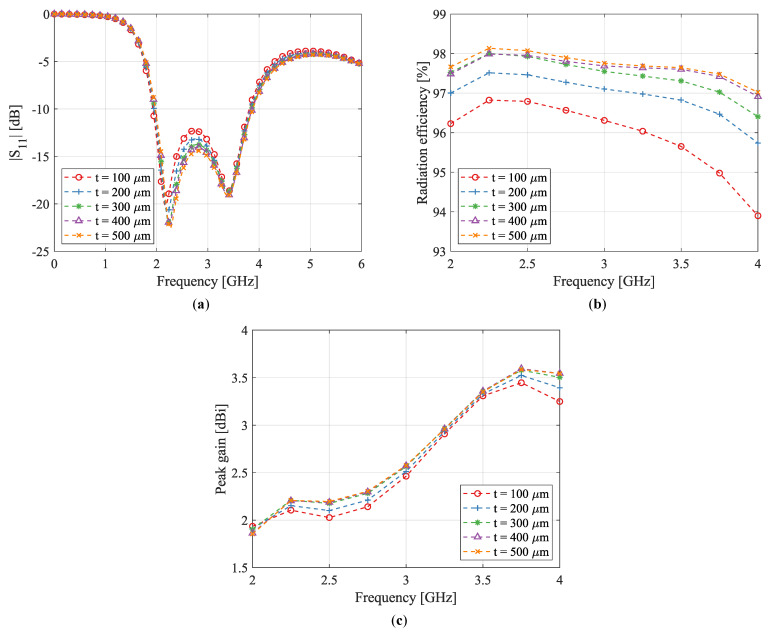
Simulated (**a**) reflection coefficient $|S_{11}|$ (in dB), (**b**) radiation efficiency (in %), and (**c**) peak gain (in dBi) for meshed antenna P-II with varying line width *t*.

**Figure 8 sensors-22-07929-f008:**
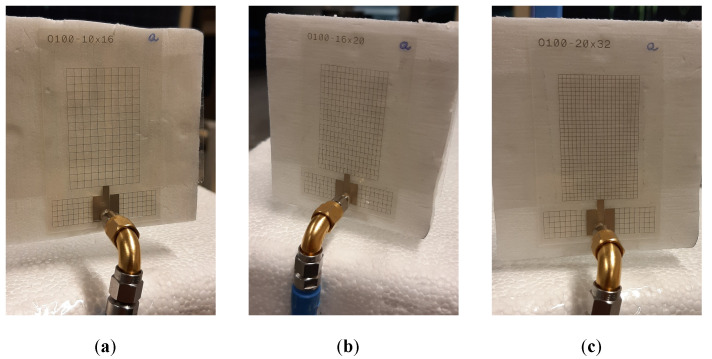
Photographs of the fabricated meshed antenna prototypes with different ($N_{p,x}\times N_{p,y}$) mesh grids: (**a**) P-II, with a (10 × 16) grid; (**b**) P-III, with a (16 × 20) grid; and (**c**) P-IV, with a (20 × 32) grid. As the antennas are largely transparent, they are shown with white styrofoam in the background.

**Figure 9 sensors-22-07929-f009:**
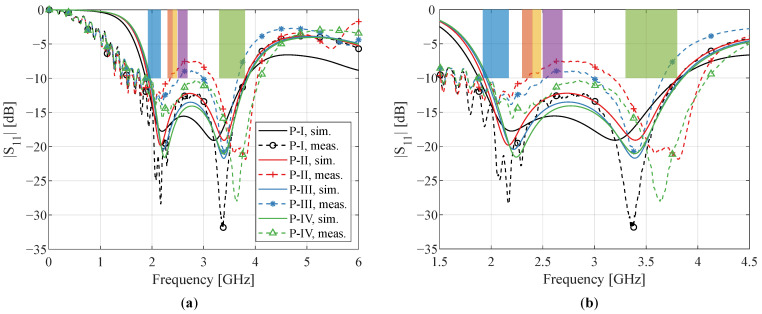
Simulated (solid lines) and measured (dashed lines) reflection coefficient $|S_{11}|$ (in dB) for the different prototypes P-I, P-II, P-III, and P-IV (**a**) in the frequency range from 0 to 6 GHz and (**b**) zoomed in to the frequency range from 1.5 to 4.5 GHz. The relevant frequency bands, namely, IMT/LTE-1/n1 (1.92–2.17 GHz), LTE-40/n40 (2.3–2.4 GHz), 2.45 GHz ISM (2.4–2.4835 GHz), IMT-E/LTE-7/n7 (2.5–2.69 GHz), and n78 5G (3.3–3.8 GHz), are indicated in blue, orange, yellow, purple, and green, respectively.

**Figure 10 sensors-22-07929-f010:**
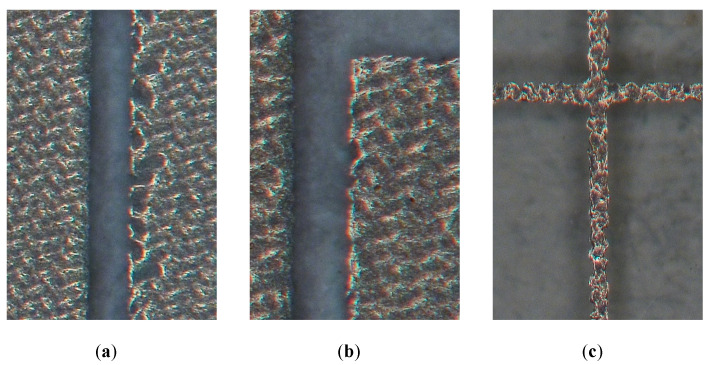
Microscopic images of the CPW feed section with nominal gap width g=200 μm (see Figure 5) for (**a**) P-I and (**b**) P-II; (**c**) mesh line with nominal width t=100 μm (see Figure 6) for P-II.

**Figure 11 sensors-22-07929-f011:**
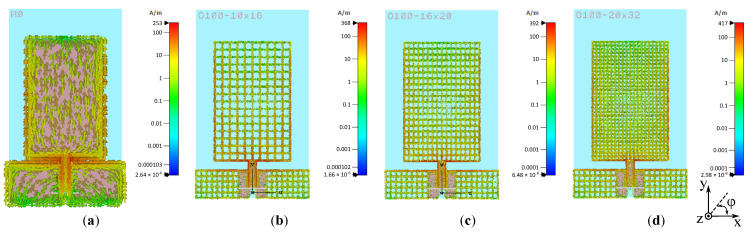
Simulated surface currents at 2.45 GHz for prototypes (**a**) P-I, (**b**) P-II, (**c**) P-III, and (**d**) P-IV.

**Figure 12 sensors-22-07929-f012:**
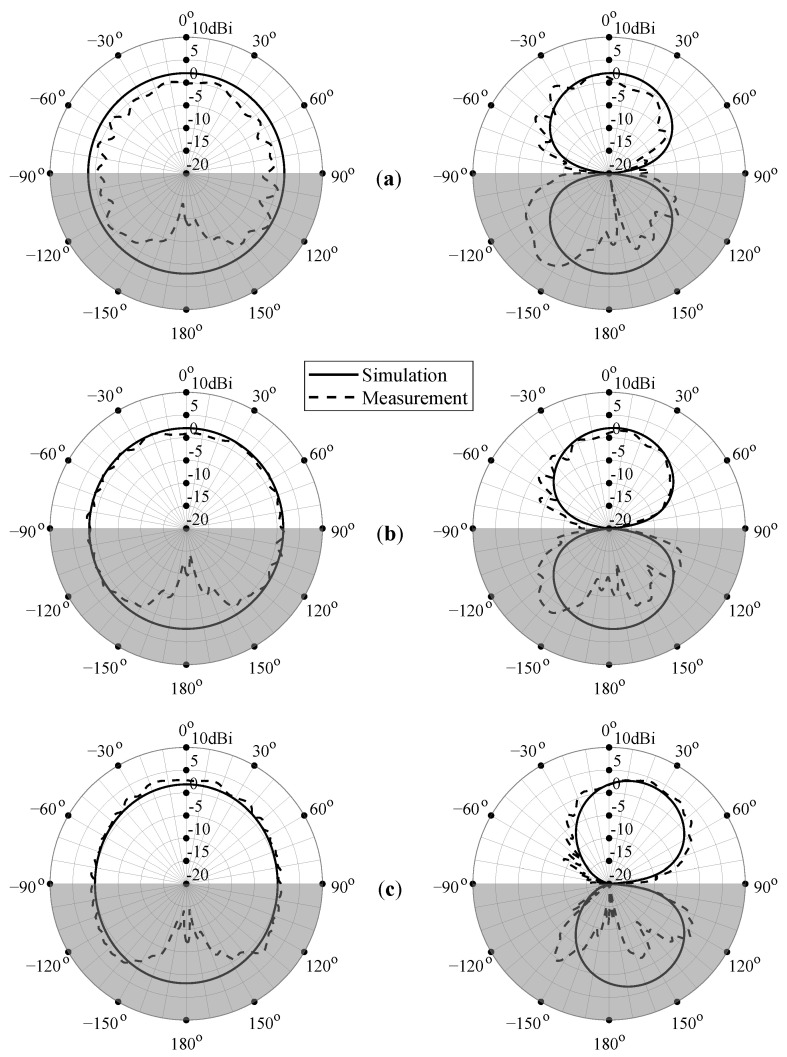
Simulated (solid lines) and measured (dashed lines) radiation pattern for P-I at (**a**) 2.05 GHz, (**b**) 2.45 GHz, and (**c**) 3.55 GHz. Left: XZ-plane; right: YZ-plane. To indicate the unreliability of the measured results around 180° due to the impact of the measurement setup, this region is shaded in gray.

**Figure 13 sensors-22-07929-f013:**
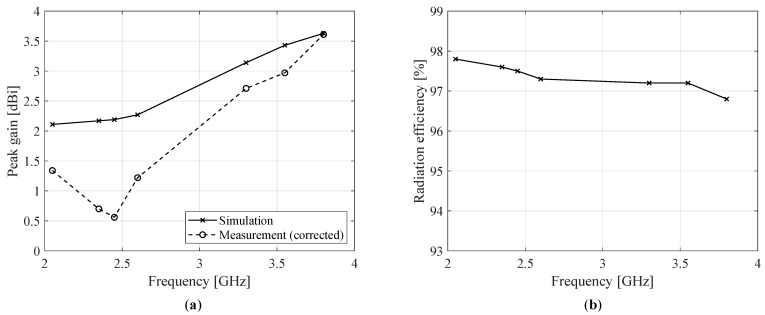
Radiation parameters for P-I: (**a**) simulated (solid lines) and measured (dashed lines) peak gain and (**b**) simulated radiation efficiency. The measured peak gain was corrected by characterizing the magnitude of the radiation pattern’s ripple through simulation of the non-idealities in the measurement setup.

**Figure 14 sensors-22-07929-f014:**
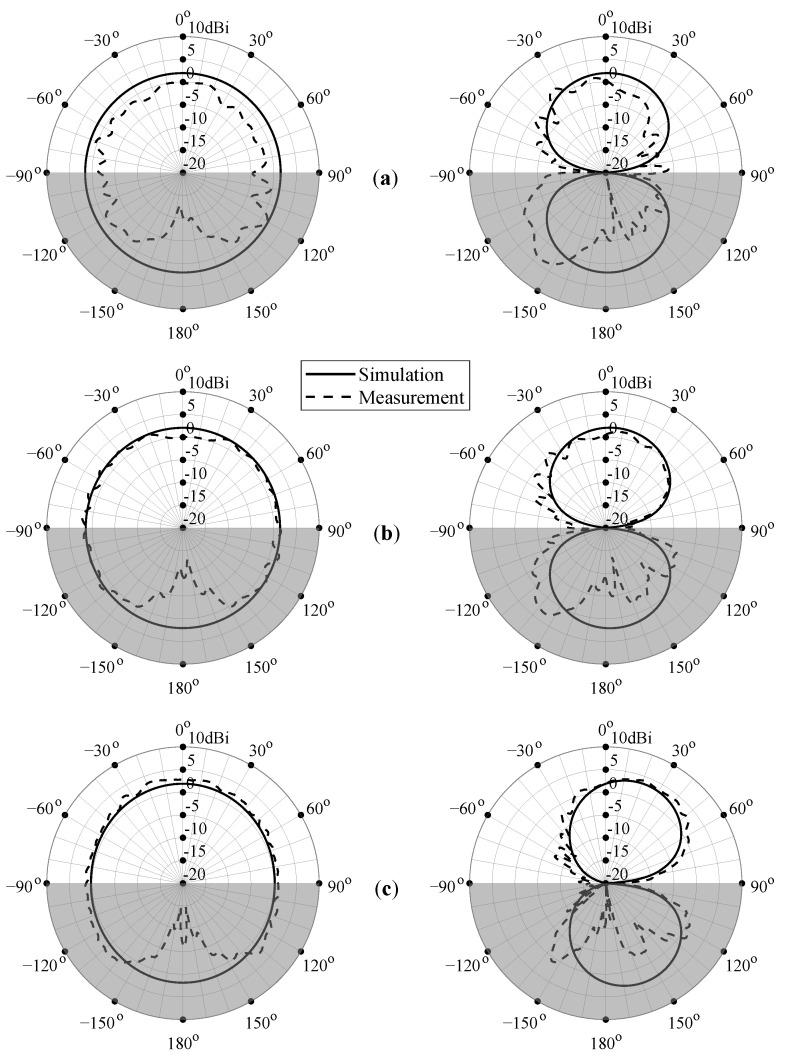
Simulated (solid lines) and measured (dashed lines) radiation pattern for P-IV at (**a**) 2.05 GHz, (**b**) 2.45 GHz, and (**c**) 3.55 GHz. Left: XZ-plane; right: YZ-plane. To indicate the unreliability of the measured results around 180° due to the impact of the measurement setup, this region is shaded in gray.

**Figure 15 sensors-22-07929-f015:**
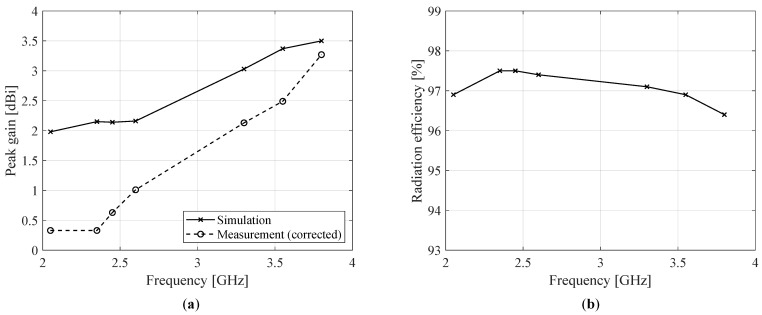
Radiation parameters for P-IV: (**a**) simulated (solid lines) and measured (dashed lines) peak gain and (**b**) simulated radiation efficiency. The measured peak gain was corrected by characterizing the magnitude of the radiation pattern’s ripple through simulation of the non-idealities in the measurement setup.

**Table 1 sensors-22-07929-t001:** Mesh parameters for P-II, P-III, and P-IV. For all prototypes, D=3.21mm, $w_{g}=1.52\mathrm{mm}$, and $l_{g}=1.58\mathrm{mm}$.

Antenna Prototype	No. of Ground Holes $\left(N_{g,x}\times N_{g,y}\right)$	No. of Patch Holes $\left(N_{p,x}\times N_{p,y}\right)$	Patch Hole Width $w_{p}$ [mm]	Patch Hole Length $l_{p}$ [mm]
P-II	(8 × 5)	(10 × 16)	2.22	2.18
P-III	(8 × 5)	(16 × 20)	1.35	1.72
P-IV	(8 × 5)	(20 × 32)	1.06	1.04

**Table 2 sensors-22-07929-t002:** Overview of the fabricated antennas.

Antenna Prototype	Dimensions [mm]	Ink Surface Area [cm^2^]	Ink/Total Surface Ratio [%]	Ink Reduction (ζ×100) * [%]	Covered Frequencies [GHz] ^†^
P-I	36.10 × 61.80	12.97	58.1	-	1.75–3.81
P-II	34.76 × 58.03	1.77	8.8	85.8	1.75–2.35 3.18–4.04
P-III	34.76 × 58.03	2.07	10.3	83.2	1.88–2.53 2.99–3.66
P-IV	34.76 × 58.03	2.46	12.2	79.8	1.88–4.10

* The relative ink reduction ζ is calculated using Equation (4). ^†^ The listed frequencies are measured values.

**Table 3 sensors-22-07929-t003:** Comparison with state-of-the-art CPW-fed printed wideband antennas.

Ref.	Printing Technology	Substrate Type	Dimensions [mm]	Dimensions w.r.t. $\lambda_{c}$ [-] ^*^	Covered Frequencies [GHz]	Peak Rad. Eff. [%]	Peak Gain [dBi]	Ink Surface Area [cm^2^] ^†^	Ink/Total Surface Ratio [%]
[5]	Inkjet printing	Ceramic	66.00 × 62.00	0.60 × 0.56 1.29 × 1.21	2.48–2.94 (17.0%) 4.87–6.82 (33.4%)	88 86	5.2 4.6	24.93	60.9
[23]	Inkjet printing	PET	47.00 × 25.00	1.08 × 0.57 2.60 × 1.38	3.04–10.70 (111.7%) 15.20–18.00 (16.9%)	98 95	4.3 5.7	5.89	50.2
[24]	Screen printing	Kapton	40.00 × 55.00	0.58 × 0.80	1.77–6.95 (118.8%)	90	5.9	9.79	44.5
[25]	Screen printing	PET	60.00 × 50.00	0.70 × 0.58	2.00–5.00 (85.7%)	86	2.0	14.26	47.5
**This work ^‡^**	**Screen printing**	**PET**	**34.76 × 58.03**	**0.35 × 0.58**	**1.88–4.10 (74.2%)**	**97.5**	**3.3**	**2.46**	**12.2**

* The dimensions are calculated with respect to the wavelength *λ_c_* at the center frequency of each supported band. ^†^ The ink surface area is estimated based on the dimensions provided in the respective papers. ^‡^ The best meshed antenna is chosen for comparison (i.e., prototype P-IV). The entry for this work is indicated in bold.

## Data Availability

Not applicable.

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
