# Peer review of "Cost-Driven Design of Printed Wideband Antennas with Reduced Silver Ink Consumption for the Internet of Things"

_sensors, 2022, doi:10.3390/s22207929_

Round 1
Reviewer 1 Report
This work prints silver-based nanoparticle ink on a polyethylene terephthalate to produce microstrip antennas. To achieve the light weight and reduce the cost, the planar parts of the antenna are meshed. Measurement results show that the designed antenna prototype perform well. Several detailed concerns are provided below.
(1) In Fig.4, why the long resonator has better S-parameter performance than the short one?
(2) The antenna is symmetrical. In Fig.9(c), why the peak radiation direction deviates from the 0o?
(3) In Fig.10, why the measured peak gain is higher than the simulated one?
Reviewer 2 Report
The paper describes an antenna design for an IoT application. The topic of the paper is clear - to design a flexible antenna with the lowest possible costs for given production technology. The paper is well organized, well written - very clear and interesting for reader.
I have a few questions / notes / recommendations to improve the manuscript (please see also the attached document with highlighted parts):
Question - at line 148/149: "...the final thickness of the deposited ink layer was approximately 4 um". How can the uniformity of the layer be ensured if the substrate surface is inhomogeneous?
Note - plaese use hard space (CTRL+Shift+Space) between value and unit (visible mistake at row 58, 83)
Note - Figure 2: could be used the same scale (for example 2:1)? (Figure 3. is on a scale approx 1:1)
Note - maybe will be better use names "Prototype I - IV" instead of "antenna with a (16-20)..." - from Table 1. is clear that the Prototype II has the coarsest mesh, Prototype IV the finest mesh (the specific number of cell is not so important in global in my mind).
Note - Table 1: could be the header of the table better described, please? The parameters description is at page 8 and it is little complicated to go back in electronic version to check the parameter. Maybe could be used full description: No. of ground holes (Ng,x Ng,y), Hole width wp [mm]... (for example).
Note - Table 2 and 3 - use a clearer mark for notes please (for example with an asterisk) - if the square is used in unit...
Please consider my comments and their implementation to the paper.
Thank you very much.

Reviewer 3 Report
In this paper, a flexible passive broadband CPW-fed monopole antenna is presented to operate in the IMT/LTE-1/n1, LTE-40/n40, 2.45 GHz ISM, IMT-E/LTE-7/n7 and n78 5G frequency bands for IoT applications. The main motivation of the study is to reduce the costs arising from the use of silver-based ink in screen-printed fabrication. For this purpose, the effect of different mesh sizes on antenna performance was investigated.
Especially in terms of the scope of new generation technologies, the paper is timely. The paper is well written; however, its organization needs to be reconsidered. In the study, first of all, a general evaluation should be made in terms of radiation performance (such as S11, gain and radiation efficiency) of antennas with different mesh structures (Prototype-I, II, III and IV). After the performance comparison, the results of the ultimate design (here in, Prototype IV) should be included. Accordingly, the organization of the paper should be reconsidered.
At the end of the introduction section, there should be a paragraph about the novelty/contribution of the paper. It should be clearly emphasized. It can also be listed as items.
The number of references is not enough. The authors should add some of the state-of-the-art (cutting-edge) studies proposed more recently (2020-2022).
Figure and table layouts need to be reviewed. For example, on page 8, the description of Figure 7 refers to Table-I. However, Table-I is on a further page (Page 12).
Figure 11 is the result of which prototype antenna. In the figure description this information (Prototype-I, II, III or IV) should be included.
In Figure 5, the coordinate system orientation should be depicted (i.e., the y-direction is unclear).
The superscripts (ie, superscripts 1 and 2) in Table 2 should be replaced with symbols to avoid confusion.
Language should be polished.
The conclusion section should be revised. The contribution of the paper to the current literature should be clearly stated comparatively.
Reviewer 4 Report
The authors presented a microstrip patch antenna on a PET substrate using conductive ink. Although such antennas have widely been studied, as indicated in the literature review as well, the results may be interesting if revised carefully. The reviewer has the following comments.
(1) The authors claim that the main contribution of this study is the ink reduction during the printing of the antenna while maintaining high radiation efficiency. Why their design gives high radiation efficiency, and which novel technique was used in the reduction of the ink? Is it the design of the antenna itself, the ink properties, or the substrate? A detailed discussion is needed in this regard.
(2) Consider improving Figures 1 and 3. Everything looks black in its current form.
(3) Show the radiation pattern of the antenna in the 360-degree range of theta (not only from -90 to 90 degrees).
(4) It is a very common phenomenon to observe the reduced measured gain and radiation efficiency compared to simulated values due to cable, substrate, etc. losses during the actual measurement. However, it is very strange to see the increased measured gain (1 dBi) compared to its simulated values at high frequencies. This behavior is consistent in this study (Fig 10 and 17). A good explanation is needed, or the authors need to check their measurement/simulation setup.
(5) The statement in the Introduction, "Frequencies for IoT can be both in sub-6 GHz and the mmWave spectrum..." needs a reference to support the argument, like, "Antenna Design for Microwave and Millimeter Wave Applications: Latest Advances and Prospects" Applied Sciences 11, no. 12: 5556; and "Integrated Microwave and mm-Wave MIMO Antenna Module with 360∘ Pattern Diversity For 5G Internet-of-Things," IEEE Internet of Things Journal, 2022.
(6) Since the antenna is aimed for IoT applications. Why this antenna is suitable for IoT? Is it because of its operating frequency, which may not be a convincing point? Or flexible behavior of the antenna? Authors are strongly encouraged to study the antenna's conformal analysis (antenna performance in different bending conditions).
Round 2
Reviewer 4 Report
The manuscript has been revised well. The authors have addressed all the comments of the reviewer. The manuscript can be accepted for publication.